# A Water-Resistant, Self-Healing Encapsulation Layer for a Stable, Implantable Wireless Antenna

**DOI:** 10.3390/polym15163391

**Published:** 2023-08-13

**Authors:** Soojung An, Hyunsang Lyu, Duhwan Seong, Hyun Yoon, In Soo Kim, Hyojin Lee, Mikyung Shin, Keum Cheol Hwang, Donghee Son

**Affiliations:** 1Department of Electrical and Computer Engineering, Sungkyunkwan University, Suwon 16419, Republic of Korea; soojung2134@gmail.com (S.A.); hslyukr@g.skku.edu (H.L.); dodoworld1993@gmail.com (D.S.); peneus577@g.skku.edu (H.Y.); 2Nanophotonics Research Center, Korea Institute of Science and Technology (KIST), Seoul 02792, Republic of Korea; isk@kist.re.kr; 3Biomaterials Research Center, Biomedical Research Institute, Korea Institute of Science and Technology (KIST), Seoul 02792, Republic of Korea; hyojinlee@kist.re.kr; 4Division of Bio-Medical Science & Technology, KIST School—Korea University of Science and Technology (UST), Seoul 02792, Republic of Korea; 5Department of Biomedical Engineering, Sungkyunkwan University (SKKU), Suwon 16419, Republic of Korea; mikyungshin@g.skku.edu; 6Department of Intelligent Precision Healthcare Convergence, Sungkyunkwan University (SKKU), Suwon 16419, Republic of Korea; 7Department of Superintelligence Engineering, Sungkyunkwan University (SKKU), Suwon 16419, Republic of Korea

**Keywords:** fluoroelastomer, implantable antenna, encapsulation, self-healing, water-resistant, wireless communication, packaging

## Abstract

Polymers for implantable devices are desirable for biomedical engineering applications. This study introduces a water-resistant, self-healing fluoroelastomer (SHFE) as an encapsulation material for antennas. The SHFE exhibits a tissue-like modulus (approximately 0.4 MPa), stretchability (at least 450%, even after self-healing in an underwater environment), self-healability, and water resistance (WVTR result: 17.8610 g m^−2^ day^−1^). Further, the SHFE is self-healing in underwater environments via dipole–dipole interactions, such that devices can be protected from the penetration of biofluids and withstand external damage. With the combination of the SHFE and antennas designed to operate inside the body, we fabricated implantable, wireless antennas that can transmit information from inside the body to a reader coil that is outside. For antennas designed considering the dielectric constant, the uniformity of the encapsulation layer is crucial. A uniform and homogeneous interface is formed by simply overlapping two films. This study demonstrated the possibility of wireless communication in vivo through experiments on rodents for 4 weeks, maintaining the maximum communication distance (15 mm) without chemical or physical deformation in the SHFE layer. This study illustrates the applicability of fluoroelastomers in vivo and is expected to contribute to realizing the stable operation of high-performance implantable devices.

## 1. Introduction

The significance of disease and healthcare has increased; thus, studies on bioelectronics that monitor the bio-signals from various organs and aid treatment are underway [1,2,3,4,5,6,7,8,9,10,11,12]. Implantable devices have been utilized for precise physiological signal monitoring and effective therapeutic assistance, characterized by direct contact with the targeted organs [13,14,15,16,17,18]. However, the use of wired systems from outside the human body results in multiple problems. Factors such as inconvenience for patients and users, noise from wires, or risk of infection hinder the proper functioning of the device [19,20]. The development of wireless technologies has resolved these issues and enhanced more efficient and effective monitoring, as well as treatment strategies [21,22,23,24,25]. The wireless system was realized with the implementation of implantable devices with various functionalities, such as data acquisition and power transmission to the battery or closed-loop system in electroceuticals [26,27,28,29]. Wireless data transmission eliminates the need for wires, providing convenience for medical professionals in data collection and analyses, as well as for users in terms of mobility and ease of use [30,31]. It creates an environment of continuous monitoring even in daily life, demonstrating its potential for significant advancements in biomedical engineering.

Wireless communication systems operate by transmitting a modulated signal through a transmission antenna, receiving it from a receiving antenna, demodulating it, and decoding the modulated information. Therefore, the overall performance of the wireless communication system is directly influenced by the performance of the antenna. Although inorganic-material-based fabrication techniques enable implantable wireless systems with a high performance, it would be difficult for them to operate effectively in wet environments. In addition, the modulus of the material is not matched with that of the human tissue and cannot follow the deformation of the tissues, resulting in inflammation in the body [32,33]. Device failure owing to harsh conditions can be prevented by encapsulating the device with adequate materials. Encapsulation materials for biomedical electronics should provide mechanically and electrically stable, reliable, and long-term usable properties. To resolve this problem, various encapsulation substances that can be inserted into the body have been introduced [34,35,36,37,38]. However, their irregular and repetitive movement hinders the uniform operation of such devices for long-term use. Encapsulation layers made of conventional encapsulation materials are breakable, resulting in a loss of function when external-damage-induced breaks are applied [39,40,41,42,43]. The devices may be damaged owing to the penetration of bodily fluid. Moreover, materials that can be swollen or influenced by moisture are not suitable for long-term encapsulation. Commercial encapsulation materials, such as syringe, ultraviolet-curable, drop-casting resin or gel types, can result in an un-uniform surface, which can deform or crack the encapsulated material. Furthermore, encapsulation can be time-consuming and complex, hindering widespread adoption. The use of commercially available encapsulation materials is characterized by complex encapsulation processes, such as mixing or drying [44,45,46]. Uniform film-type encapsulation with a waterproofing effect and self-bonding properties that can be easily applied to resolve these problems is required.

Here, we introduce a water-resistant, self-healing encapsulation layer for a stable, implantable wireless antenna (Figure 1a). First, we designed reader and tag antennas that could transmit information from the subcutaneous domain to the outside. Second, we encapsulated the antenna with an autonomous, self-healing fluoroelastomer (SHFE) to ensure that the antenna was protected and maintain its intended purpose in the long term. Research related to self-healing materials has been actively reported, covering various mechanisms and their applications [47,48,49,50,51]. Among the various types of self-healing materials, we opted for a strategy utilizing the SHFE via dipole–dipole interaction. The material has been reported as being stretchable and autonomously self-healable at room temperature, even underwater, like the characteristics of human tissues [49]. To realize these self-healing properties and contain a higher dipole moment, PVDF-HFP containing more than 40% HFP was blended with DBP. Due to fluorine (F) having a higher electronegativity compared to carbon (C), a dipole was formed in the CF_3_ group of the PVDF-HFP. These formed dipoles interacted with adjacent CF_3_ dipoles, leading to intermolecular interactions that imparted self-healing properties to the material. Furthermore, due to its low T_g_ and the help of DBP serving as a plasticizer, an enhanced polymer mobility led to improved intermolecular interactions, thereby strengthening the self-healing properties without external agents or a trigger such as heat or light. [49]. Additionally, the interference of water molecules could be prevented by not utilizing metal cations or ionic components. Owing to its self-healing properties, facile encapsulation is achievable using a film-type SHFE. The film fabricated using this method is not simply sticky, but it autonomously makes a seamless and homogeneous interface as if it were one. Moreover, self-healing properties enable the self-recovering of polymers even when they are damaged by external forces and prevent bodily fluid penetration. The SHFE is highly hydrophobic, chemically stable, and has similar mechanical properties to tissues. We fabricated an implantable wireless antenna using the SHFE by placing an antenna between two films (Figure 1b). The performance of the antenna was validated under porcine skin ex vivo. Finally, we wirelessly measured the maximum communication distance in the rodent’s back for four weeks to validate its performance as an encapsulation material and implantable wireless antenna.

## 2. Materials and Methods

### 2.1. Fabrication of the SHFE Film

All the solvents were purchased from Sigma-Aldrich (Burlington, MA, USA). The SHFE films were prepared by dissolving 5.1 g of PVDF-HFP (3M™ Dyneon™ Fluoroelastomer FC 1650, 3M Co., Ltd., St. Paul, MN, USA) and 900 mg of DBP (Sigma-Aldrich, USA) in 30 mL of anhydrous acetone (Sigma-Aldrich, USA) and stirring at room temperature for 3 h. The mold for film formation employed polyethylene. Initially, a square glass of 10 cm × 10 cm was prepared. Double-sided tape was applied to one side of the glass entirely. A polyethylene bag was cut into a size of 12 cm × 12 cm. The cut polyethylene bag was aligned to the center of the glass. The corners were folded and secured with tape to form the mold. The inside of the mold was cleaned with ethanol. Next, the solution was poured into a polyethylene mold and dried at room temperature overnight to remove the solvent residue. Upon drying and the evaporation of the solvents, the SHFE film was delicately peeled off and placed into a polyethylene bag. Any air bubbles that may have formed during the transfer were carefully pushed out and removed.

In the case of colored films, which were produced to clearly illustrate the self-healing aspect, marker pens were dissolved in the solvent to create the desired colors (red and blue). This involved a process of dissolving the non-permanent marker in the solvent and allowing the dye in the ink to mix thoroughly with the solvent, thereby producing a colored film when the solution was applied and dried.

### 2.2. Mechanical Characteristics of the SHFE

A universal testing machine (UTM, Instron 5565, Norwood, MN, USA) was used to measure the mechanical properties. SHFE samples with a 5 mm width and 20 mm length were prepared. Once the samples were prepared, they were loaded into the UTM with an initial length of 5 mm. The speed was 20 mm min^−1^. Throughout the experiment, all the measurement data were collected using the Instron Bluehill software. For the evaluation of the intrinsic self-healing capabilities, the sample was bisected lengthwise and the cut surfaces were placed together underwater overnight.

### 2.3. Measurement of Water Vapor Transmission

The water vapor transmission rate was measured at 37.8 degC and 100% humidity using Aquatran 2 (Mocon, Inc., Minneapolis, MN, USA) with a detection limit of 5 × 10^−5^ g m^−2^ per day. SHFE films were prepared with an area of 9 cm × 9 cm. PDMS–MPU_0.4_–IU_0.6_ self-healing polymer (SHP) films were used as controls. The PDMS–MPU_0.4_–IU_0.6_ SHPs were fabricated using a previously reported synthesis procedure by regulating the molar ratios of 4,4′-methylenebis(phenylurea) (MPU) and isophorone bisurea (IU) to polydimethylsiloxane (PDMS), with MPU and IU mixed in a 4:6 ratio [48]. The samples were fabricated with identical dimensions.

### 2.4. Cell Viability Test of the SHFE

A cell viability test was performed on the SHFE films using the primary neuron brain cell. The samples were cut into dimensions of 1.5 cm × 1.5 cm, washed twice in ethanol, and immersed in phosphate-buffered saline (PBS) for 7 days. A total of 4 × 6 experiments were conducted on three samples and one control sample 6 times. The absorbance was measured by dyeing the cell and set to an average of 1. The cell viability was evaluated by comparing the absorbance of each sample.

### 2.5. Design of an RFID Reader and Tag Antennas That Communicates via Magnetic Resonance in the Air

The geometry of the radio frequency identification (RFID) reader and tag antennas designed for the 13.56 MHz band in the air is shown in Figure 2a,b. The proposed antennas were printed on a Taconic TLY-5 substrate with a dielectric constant of 2.2 and a thickness of 0.25 mm. The size of the reader antenna was 98.4 mm × 91 mm with 7 turns, a coil line width of 2.15 mm, and a line spacing of 0.45 mm. The size of the tag antenna was 43.6 mm × 40 mm with 9 turns, a coil line width of 0.9 mm, and a line spacing of 0.5 mm. The antennas were fed using an SMA connector. The reader antenna was designed to resonate at 13.56 MHz by connecting a capacitance of 20.4 pF in series to achieve a reactance value close to 0. Additionally, an Ntag 213 chip was connected in parallel with the tag antenna, with a chip input capacitance of 50 pF. The tag antenna was designed to have an inductance value of 2.75 μH for resonance at 13.56 MHz. The simulated inductance value of the tag antenna, with the simulation performed using the 3D Computer Simulation Technology Microwave Studio, is shown in Figure 1c.

### 2.6. Implantable Wireless Antennas Packaging

The SHFE films were cut using a razor blade into sizes of 2.8 cm × 3.5 cm. One sheet of SHFE was placed on the surface and the antenna was positioned at the center. Another sheet was aligned and placed to completely overlap with the first sheet, and gentle pressure was applied. The assembly was left at room temperature overnight.

### 2.7. Ex Vivo Experimental Method of Wireless Communication Using the Implantable Antenna

The tag antenna, equipped with an NTAG213 chip and SHFE, was implanted between porcine fat and muscle. The reader antenna, fabricated for measuring the recognition distance of the implanted tag antenna, was directly connected to a commercial RFID module via an MMCX to SMA cable. This commercial RFID module was connected to a laptop and then linked with the software provided by the module manufacturer. When a signal was transmitted to the reader antenna, the recognition distance was measured based on whether the unique ID of the NTAG213 chip was identified through the transmitted signal. The measurement of the recognition distance was not taken from the position where the tag was implanted, but from the skin. For precise measurements, multiple styrofoam boards of 1 mm and 3 mm with various combinations were utilized to measure the maximum recognition distance.

### 2.8. In Vivo Experimental Method of Wireless Communication Using the Implantable Antenna

An in vivo experiment for wireless communication was conducted to demonstrate the feasibility of the implantable antenna. The SHFE films for encapsulation were fabricated considering the size of the antenna, resulting in dimensions of 1.3 cm by 1.7 cm. The fabricated antenna was positioned between two sheets of the prepared film and then left to stand overnight at room temperature. For sterilization, the samples were exposed to UV for 2 h on the front and rear sides (1 h each). Animal experiments were conducted on rodents (Sprague–Dawley rats, 8 weeks old). The rats were anesthetized by inhalation of 1.5% isoflurane with oxygen. The backs of the rats were shaved and sterilized. Subsequently, an incision of approximately 4 cm was made on the back region using surgical scissors. After securing space between the subcutaneous layer and the muscle, the sterilized antenna was inserted. The incision site was then sutured using black silk sutures. The experimental site was disinfected to finalize the implantation process. The experiment was conducted over a span of 4 weeks. To ensure precise measurements, just as in the ex vivo experiment, a board was assembled using multiple 1 mm and 3 mm styrofoam pieces (up to 15 mm in total), which was then attached to the reader antenna for the distance measurements. By affixing the styrofoam to the reader antenna, it was possible to confirm that the maximum distance could be stably maintained and measured, even with freely moving rats.

### 2.9. H&E Staining

The tissues, which were directly in contact with the implantable antenna encapsulated with the SHFE, were collected. The collected tissues were gently rinsed in PBS and fixed with 4% paraformaldehyde for 24 h at room temperature. The specimens were sectioned and stained with hematoxylin and eosin (H&E) in GENOSS (Yeongtong-gu, Suwon-si, Republic of Korea). The H&E staining analysis was conducted using a decalcification method for the sham, SHFE, and bare groups and an undecalcification method for the commercial epoxy group. All sections were visualized using a digital-slide-visualizing program (Caseviewer, 3DHISTECH Ltd., Budapest, Öv u. 3., Hungary).

## 3. Results

### 3.1. Mechanical Properties of the SHFE

We evaluated the mechanical properties of the PVDF-HFP film according to the change in the amount of DBP by pulling the samples using the UTM (Figure 3a). We increased the weight ratio of the DBP from 5% to 15%. Considering the requirement for shape retention, we chose the softest material that still allowed for the shape maintenance of the SHFE, and it contained 15% DBP. The Young’s modulus of the SHFE film was calculated to be approximately 0.4 MPa. Next, the contact angle was measured to validate the water resistance (Figure 3b). For the evaluation of the self-healing properties, the SHFE films were completely cut with a razor blade, overlapped, healed at room temperature underwater and overnight, and then stretched (Figure 3c). In addition, for the self-healed SHFE, the films were cut and the cut surfaces were reattached as they were. The stretchability was maintained at a level of at least 450%, even after self-healing (Figure 3d). Next, the water vapor transmission ratio (WVTR) was followed to evaluate the barrier hermeticity (Figure 3e). The result of the WVTR for the SHFE was 17.8610 g m^−2^ day^−1^, whereas it was 148.9983 g m^−2^ day^−1^ for the PDMS–MPU_0.4_–IU_0.6_ SHP. It was confirmed that the SHFE demonstrated a significantly higher water resistance compared to the PDMS–MPU_0.4_–IU_0.6_ SHP. Finally, a cell viability test was performed on a primary neuron cell to validate the biocompatibility. The biocompatibility was validated by demonstrating that the cell viability of the primary neuron cells was similar to that of the control group (Figure 3f).

### 3.2. Measurement of the RFID Reader and Tag Antennas in the Air

The Proxmark3 DEV kit and RFID module used to measure the recognition distance of the fabricated reader and tag antennas are shown in Figure 4a. The commercial reader and tag antenna from left to right, followed by the proposed reader and tag antenna, are shown in Figure 4b. The environment for measuring the recognition distance between the reader and tag antennas is shown in Figure 4c. The recognition distances between the reader and tag antennas, as measured using a fabricated styrofoam jig, are listed in Table 1. Based on the measurement results, the proposed magnetic resonance-based reader and tag antennas could be recognized from a longer distance compared to the commercial reader and tag antennas.

### 3.3. Measurement of the RFID Tag Antenna Inserted under the Porcine Skin

The geometry of the tag antenna encapsulated with the SHFE is shown in Figure 5a. Porcine tissue has similar electrical properties to human tissue [52]; at 13.56 MHz, the dielectric constants of the porcine skin, fat, and muscle are 380, 40, and 105, respectively [53]. The previously designed tag antenna was tuned considering its insertion into porcine tissue and a tag antenna with an inductance value of 2.75 μH was fabricated to resonate at 13.56 MHz, considering the 50 pF capacitance of the Ntag 213 chip. When the tag antenna was inserted into the porcine tissue, which has similar properties to that of human tissue, the chip stopped functioning, owing to shorts caused by moisture. The entire tag antenna was encapsulated with the SHFE in order to cause the chip to function. The measured inductance value of the tag antenna under the porcine skin with no additional chip is shown in Figure 5b. The measured inductance value was 2.1545 μH. To resonate at 13.56 MHz, a chip with a capacitance value of 64 pF was required, based on the resonance condition $2\pi f_{0}=1/\sqrt{L\times C}$. Because the input capacitance value of the Ntag 213 chip in the tag antenna was 50 pF, a 14 pF chip was added by soldering it to the tag. As shown in Figure 5c, the remeasured inductance value was obtained as 2.7522 μH. The environment for measuring the recognition distance between the tag inserted into the porcine model and the reader is shown in Figure 5d. The stored information on the tag antenna chip was recognized at a maximum distance of 18 mm using a reader antenna that resonated at 13.56 MHz in the air. The maximum recognition distance was 20% shorter than that measured between the reader and tag antennas in the air, owing to the difference in electrical properties between the air and porcine tissue.

### 3.4. Wireless Communication In Vivo Using SHFE-Encapsulated Implantable Antenna

When foreign substances enter the body, side effects such as toxicity, immune, inflammation, and allergic reactions can occur, which can destroy the structure and function of the body cells and tissues, as well as change the tissue thickness. These alterations may influence the efficiency of antennas that are designed to satisfy specific performance criteria. Our encapsulation layer, owing to its low Young’s modulus (~0.4 MP), stretchability, biocompatibility, and water resistance, minimized this impact on tissues. Based on the nature of the SHFE, we assumed that it showed little tissue damage and maintained the maximum communication distance of the wireless antenna. We validated this by measuring the maximum communication distance as a comparison group with a bare antenna designed to be operable in the body, a sample wrapped with the SHFE, and a sample encapsulated using a general commercial epoxy. Smith charts in the air and under the skin of the rodent are shown in Figure 6a,b, respectively. The wireless communication in vivo environment is described in Figure 6c. On day 0, the bare antenna did not work, owing to a short circuit in the bodily fluid environment (Figure 6d). A maximum communication distance of 15 mm was validated for the antennas with SHFE encapsulation and those surrounded by commercial epoxy. The experiment was conducted for four weeks. The commercial antennas suddenly stopped functioning in the midst of the experiment, whereas the antennas wrapped with the SHFE maintained a maximum distance of 15 mm throughout the entire period (Figure 6c,d). After the measurement was completed, the stain on the antenna wrapped with the SHFE clearly disappeared after it was slightly rinsed in PBS (Figure 6e). This indicated that the SHFE is water resistant and chemically stable. In addition, H&E staining was performed on the tissues where the sample was inserted. Severe inflammatory reactions were observed in the commercial epoxy and non-encapsulated samples. However, the SHFE exhibited significantly lower inflammatory reactions compared to those of the commercial epoxy and non-encapsulated sample (Figure 6f).

## 4. Discussion

For antennas implanted within the body, the environment for antenna insertion varies even within similar body parts due to factors such as the density of the muscles and the thickness of the fat, which differ among individuals. As such, impedance matching tailored to each circumstance is required. The fabricated tag antenna undergoes an analysis of the insertion site’s environment and subsequent impedance matching prior to its implantation. Furthermore, impedance matching can be conducted even when the internal environment changes due to disease, weight gain, or loss during insertion. This suggests the potential applicability of this study across various fields requiring diverse information from implanted devices.

Via the facile and stable encapsulation of antennas using the SHFE, we realized implantable wireless antenna communications. Despite the antenna being made of a sturdy and sharp-edged substrate, the implantable antenna wrapped with the SHFE rarely caused tissue changes, owing to the tissue-like modulus and water resistance of the SHFE films. These properties of the SHFE films not only ensured minimal interaction with the surrounding tissues, but also prevented potential tissue damage from the sharp edges of the substrate. It maintained the intended performance of the implantable wireless antenna for four weeks in vivo by preventing biofluid penetration, toxic reactions, and tissue deformation. As implantable devices trend towards increased complexity and miniaturization, these SHFE films are anticipated to be broadly and usefully applied to ensure device performance while protecting these devices within the body.

## 5. Conclusions

We designed an implantable wireless communication antenna for operation within the subcutaneous region of a rodent. Its stable performance was achieved by combining an antenna specifically tuned for in-body communication with a water-resistant SHFE used as an encapsulation layer. The antenna for subcutaneous use in rodents had an inductance of L = 2.2115 μH and was tuned for the 13.56 MHz band. Using an NTAG213 chip with a 12 pF capacitance, it achieved a resonant inductance of L = 2.7506 μH. Furthermore, to validate the performance of the SHFE as an encapsulator, we evaluated its Young’s modulus (approximately 0.4 MPa), self-healing efficiency (at least 450%, even after self-healing in an underwater environment), contact angle (average 99.4°), WVTR (WVTR result: 17.8610 g m^−2^ day^−1^), and cell viability. By simply applying SHFE encapsulation via a film-to-film self-bonding process, we confirmed its stable operation at a maximum distance (15 mm) within the subcutaneous region of a rat for 4 weeks. The water-resistant characteristic ensured that the samples retrieved after the in vivo experiments showed any internal damage or bodily fluid penetration. Finally, the histology results showed that the soft and biocompatible SHFE encapsulation layer was capable of minimizing skin inflammatory responses and alleviated the potential skin deformations caused by rigid materials.

## Figures and Tables

**Figure 1 polymers-15-03391-f001:**
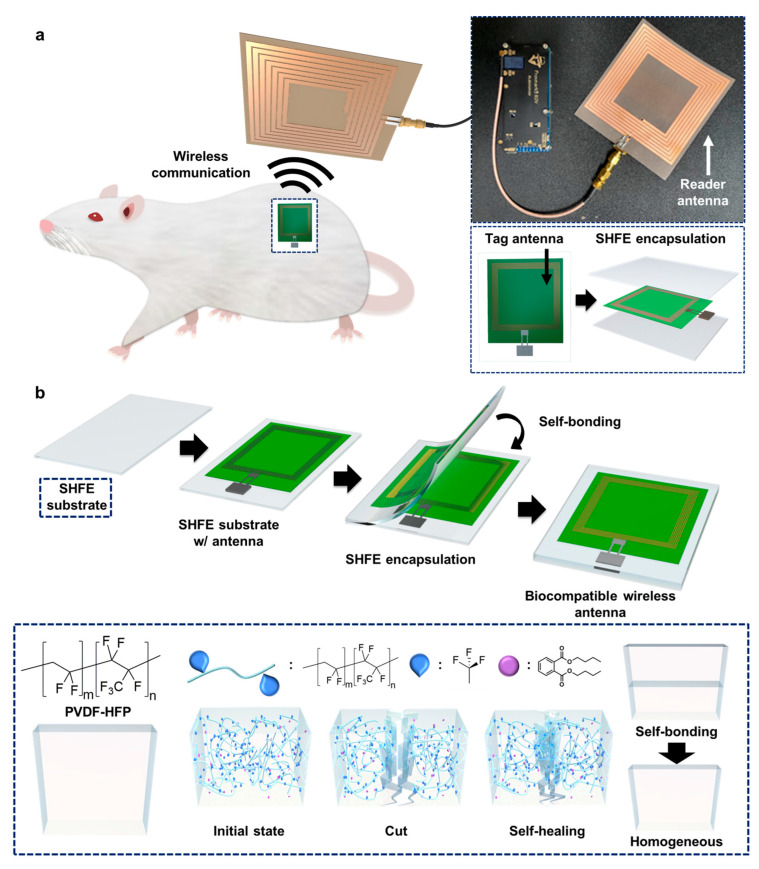
Overall concept of wireless antenna communication in vivo via SHFE film encapsulation. (**a**) Schematic of wireless antenna communications of implantable antenna with SHFE film encapsulation. (**b**) Schematic of facile and stable SHFE encapsulation of tag antenna (top) and illustration of molecular design and self-healing mechanism of SHFE (bottom).

**Figure 2 polymers-15-03391-f002:**
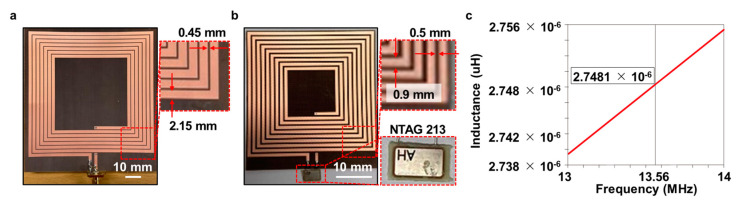
Design and simulation result of magnetic resonance−based RFID reader and tag antennas: (**a**) proposed reader antenna; (**b**) proposed tag antenna; and (**c**) simulated inductance versus frequency of the tag antenna.

**Figure 3 polymers-15-03391-f003:**
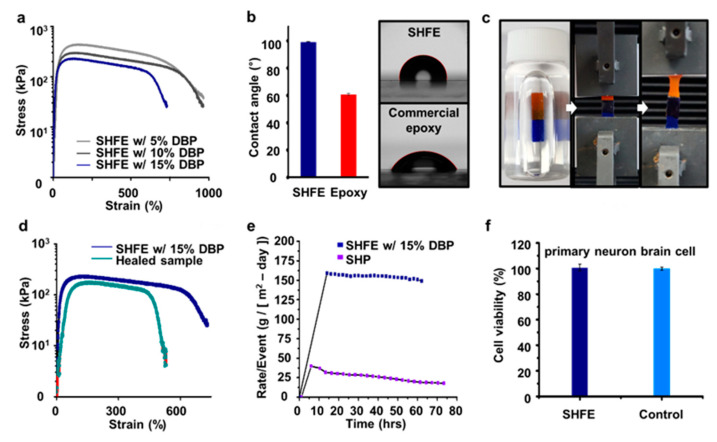
Mechanical properties of the SHFE. (**a**) Stress–strain curves of SHFE films with DBP molar ratios of 5%, 10%, and 15%. (**b**) Contact angle of deionized water on SHFE and commercial epoxy film. (**c**) Underwater self-healability of overlapped SHFE. (**d**) Stress–strain curves of reattached self-healed SHFE films after razor blade cut. (**e**) WVTR result of SHFE. (**f**) Statistical analysis of cell viability.

**Figure 4 polymers-15-03391-f004:**
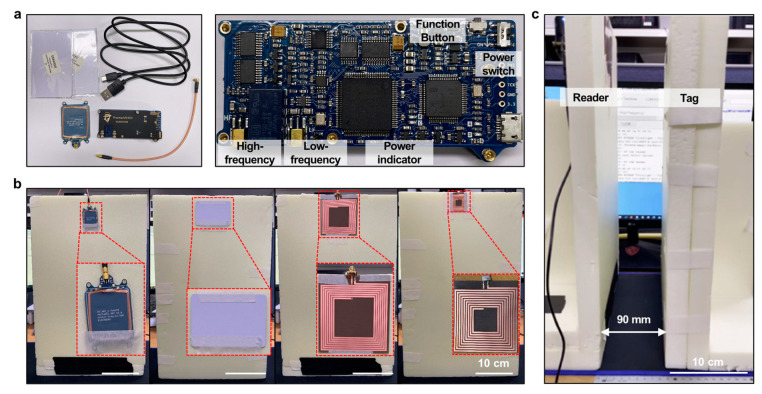
Measurement of air-gap recognition distance in the magnetic resonance-based RFID reader and tag antennas: (**a**) from left to right, Proxmark3 DEV kit and Proxmark3 module. (**b**) From left to right, commercial reader, tag antenna, proposed reader, and tag antenna. (**c**) Measurement environment for the recognition distance between the proposed reader and tag antennas.

**Figure 5 polymers-15-03391-f005:**
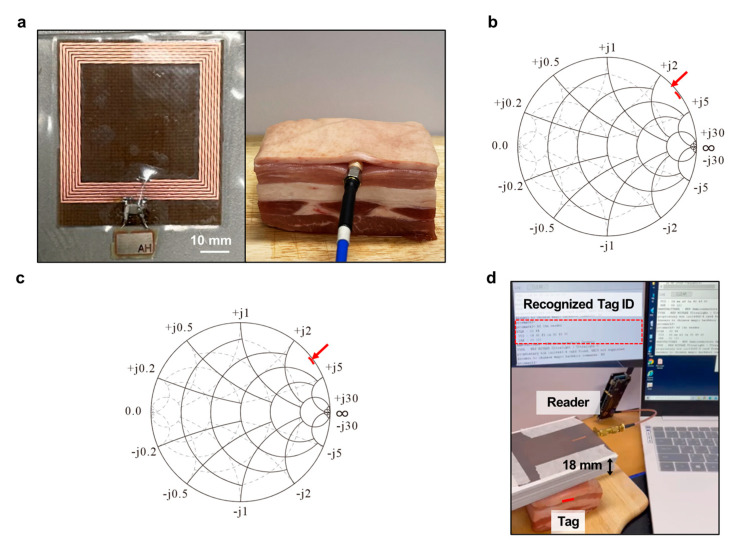
Design and measurement of the RFID tag antenna inserted inside the porcine model ex vivo: (**a**) Polymer−encapsulated tag antenna configuration, tag antenna inserted into the porcine model. (**b**) Measured inductance value of the tag antenna in the porcine model with no additional chip. Red arrow indicates the resonance frequency of 13.56 MHz; impedance of 12.773 + j234.49; converted reactance X of 2.7522 μH. (**c**) Measured inductance value of the tag antenna in the porcine model with an additional chip. Red arrow indicates the resonance frequency of 13.56 MHz; impedance of 5.6869 + j183.56; converted reactance X of 2.1545 μH. (**d**) Measurement environment for the recognition distance between the tag and reader antennas inserted into the porcine model.

**Figure 6 polymers-15-03391-f006:**
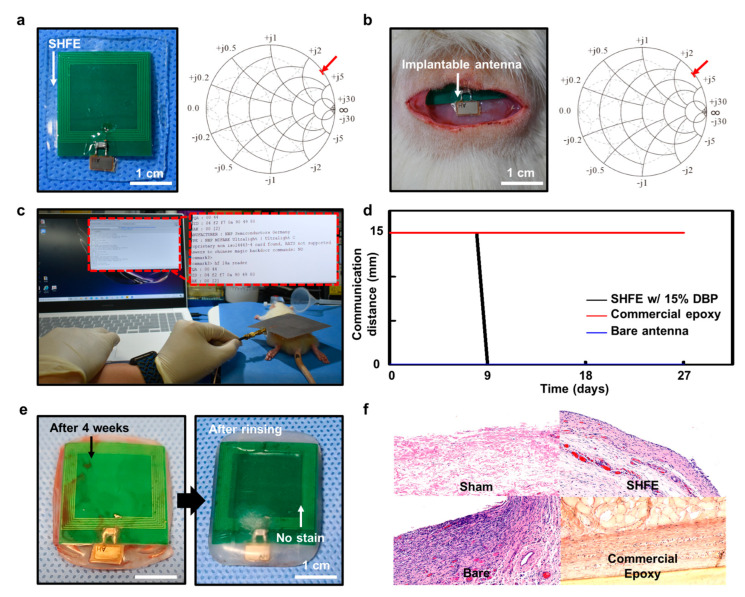
Wireless communication using SHFE−encapsulated implantable antenna inserted inside the rat model in vivo: (**a**) Implantable antenna with the SHFE and its Smith’s chart in the air. Red arrow indicates the resonance frequency of 13.56 MHz; impedance of 18.56 + j234.35; converted reactance X of 2.7506 μH. (**b**) Implanted antenna with the SHFE and its Smith’s chart under the skin of a rodent. Red arrow indicates the resonance frequency of 13.56 MHz; impedance of 9.3781 + j188.42; converted reactance X of 2.2115 μH. (**c**) Wireless communication in vivo. (**d**) Graph of the maximum distance of the implantable wireless antenna from day 0 to 27. (**e**) Implanted antenna slightly rinsed in PBS. (**f**) H&E staining images of rodent tissue.

**Table 1 polymers-15-03391-t001:** Maximum recognition distance between measured commercial readers and tags and proposed readers and tags.

	Commercial Reader Antenna	Proposed Reader Antenna
**Commercial tag antenna**	75 mm	50 mm
**Proposed tag antenna**	70 mm	90 mm

## Data Availability

The data presented in this study are available on request from the corresponding author.

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
