# Peer review of "A Water-Resistant, Self-Healing Encapsulation Layer for a Stable, Implantable Wireless Antenna"

_polymers, 2023, doi:10.3390/polym15163391_

Round 1
Reviewer 1 Report
The authors fabricated implantable wireless antennas by combating self-healing fluoroelastomer (SHFE) and designed antennas, and studied the wireless communication in vivo through experiments. The manuscript was given a title: Water-resistant self-healing encapsulation layer for stable implantable wireless antenna. But there is less content of self-healing encapsulation layer materials. Here are the comments for the authors.
1. Authors should provide a title that is highly relevant to the content of the study.
2. “polymerdibutyl phthalate (DBP) as plasticizers” in Page 2, line 87 was inconsistent with “dibutyl phthalate (DBP)” in Page 3, line 106.
3. How are implantable wireless antennas packaged? The author should add detailed methods.
4. Reference 4 does not cover the synthesis and properties of PDMS–MPU0.4–IU0.6 polymer. Why was this polymer chosen as a control? I think the author may be referring to this literature, “Jiheong Kang, Adv. Mater. 2018, 1706846”.
5. There is less data on the performance of self-healing materials. The authors designed three polymer systems containing different content of DBP. What about DBP -free luoroelastomer? Why 15% DBP were chosen to use?
6. The authors chose to study the self-healing properties at room temperature. In fact, the temperature in the animal is higher than the room temperature, what is the effect of temperature on the self-healing properties of the material? How do authors evaluate the self-healing performance of packaged antennas? I recommend supplementing the relevant data.
Author Response
Thank you for your thoughtful comments on our paper, ‘Water-resistant self-healing encapsulation layer for stable implantable wireless antenna’, submitted for publication in Polymers. Please find our explanations below, which hopefully address the issues presented, and they are revised in newly submitted version of the manuscript. The changes made in the manuscript are highlighted in yellow so that they can be easily traced.

Reviewer 2 Report
An and co-workers describe the water-resistant self-healing fluoroelastomer (SHFE) as an encapsulation material for antennas. The design of the manuscript is OK. However, the manuscript can be published after a major revision.
1. Please include a summary of the quantitative results obtained in the abstract section to provide a concise overview of the findings.
2. The literature review is weak. Please review other papers published in this field and highlight the innovation of the current research.
3. The figures in Figure 3 are not properly referenced in the text. Additionally, these results are only reported and not analyzed or discussed.
4. Please add a conclusion section to the text and summarize the results of this study.
5. The references to all figures in the text are incorrect, and one of the figures is not titled, which can confuse the reader.
To continue the review, it is necessary to correct the figure numbers and references in the text.
Author Response

(The authors gave the same response as above.)

Round 2
Reviewer 1 Report
The authors have satisfactorily addressed all my comments, and I think it can be accepted for publication in its present state.
Author Response
We appreciate your constructive and positive comments.
Reviewer 2 Report
The efforts of the authors to prepare the revised file are appreciated. Some issues have been corrected, but some other corrections are still needed.
1- The answer to the second comment has not been added to the manuscript.
2- You have mentioned the self-healing properties of the introduced polymer in several places in the manuscript. What is your definition of self-healing? Generally self-healing in polymers refers to the ability of a material to repair itself after being damaged or broken. This can occur through various mechanisms, such as the release of healing agents or the reformation of chemical bonds within the material. Which test has proven the self-healing properties of your polymer? Explain this item completely or if you do not have a convincing explanation, remove it from the entire manuscript.
3- The results section looks more like an advertising brochure than a scientific paper. In this section, you should avoid bringing up the topics that you mentioned once in the "introduction" or "materials and methods"sections and you should only report the results of your tests and fully describe the parameters that can be extracted from each test. Therefore, delete any content that is not relevant to the report of test results (lines 213-236).
4- Extract all mechanical parameters (UTS, Young's modulus, yielding strength, maximum strain at break point, toughness, etc.) from the stress-strain plots and compare the obtained values for all samples.
5- The contact angle test should be performed for all samples and compared the obtained results completely.
6- Sections 3.2.-3.4.: Please clarify the relation of these sections with the designed polymer. Also, provide a mechanism to improve the mentioned property.
7- The conclusion section is very similar to the introduction section. In this section, you should summarize the results of the tests and provide a summary of the mechanism for improving the properties.
7-
Author Response
We appreciate the reviewers for insightful comments and suggestions. We have tried to address all the feedbacks in a point-by-point manner in the response letter. The modifications are highlighted in yellow in the revised manuscript.

Round 3
Reviewer 2 Report
The manuscript can be accepted in present form